# Biodiversity and Metabolic Potential of Bacteria in Bulk Soil from the Peri-Root Zone of Black Alder (*Alnus glutinosa*), Silver Birch (*Betula pendula*) and Scots Pine (*Pinus sylvestris*)

**DOI:** 10.3390/ijms23052633

**Published:** 2022-02-27

**Authors:** Anna Gałązka, Anna Marzec-Grządziel, Milan Varsadiya, Jacek Niedźwiecki, Karolina Gawryjołek, Karolina Furtak, Marcin Przybyś, Jarosław Grządziel

**Affiliations:** 1Department of Agricultural Microbiology, Institute of Soil Science and Plant Cultivation—State Research Institute, Czartoryskich St. 8, 24-100 Pulawy, Poland; agrzadziel@iung.pulawy.pl (A.M.-G.); kgaw@iung.pulawy.pl (K.G.); kfurtak@iung.pulawy.pl (K.F.); jgrzadziel@iung.pulawy.pl (J.G.); 2Department of Ecosystems Biology, University of South Bohemia in České Budějovice, Branišovská 31, 370 05 České Budějovice, Czech Republic; mvarsadiya@prf.jcu.cz; 3Department of Soil Science Erosion and Land Conservation, Institute of Soil Science and Plant Cultivation—State Research Institute, Czartoryskich 8, 24-100 Pulawy, Poland; jacekn@iung.pulawy.pl; 4Department of Plant Breeding and Biotechnology, Institute of Soil Science and Plant Cultivation—State Research Institute, Czartoryskich St. 8, 24-100 Pulawy, Poland; mprzybys@iung.pulawy.pl

**Keywords:** forest ecosystem, forest bulk soil, biological activity, 16S rRNA, next-generation sequencing (NGS), physiological profiles of soil microorganisms, Biolog EcoPlates

## Abstract

The formation of specific features of forest habitats is determined by the physical, chemical, and biological properties of the soil. The aim of the study was to determine the structural and functional biodiversity of soil microorganisms inhabiting the bulk soil from the peri-root zone of three tree species: *Alnus glutinosa*, *Betula pendula*, and *Pinus sylvestris*. Soil samples were collected from a semi-deciduous forest located in an area belonging to the Agricultural Experimental Station IUNG-PIB in Osiny, Poland. The basic chemical and biological parameters of soils were determined, as well as the structural diversity of bacteria (16S ribosomal RNA (rRNA) sequencing) and the metabolic profile of microorganisms (Biolog EcoPlates). The bulk soils collected from peri-root zone of *A. glutinosa* were characterized by the highest enzymatic activities. Moreover, the highest metabolic activities on EcoPlates were observed in bulk soil collected in the proximity of the root system the *A. glutinosa* and *B. pendula*. In turn, the bulk soil collected from peri-root zone of *P. sylvestris* had much lower biological activity and a lower metabolic potential. The most metabolized compounds were L-phenylalanine, L-asparagine, D-mannitol, and gamma-hydroxy-butyric acid. The highest values of the diversity indicators were in the soils collected in the proximity of the root system of *A. glutinosa* and *B. pendula*. The bulk soil collected from *P. sylvestris* peri-root zone was characterized by the lowest Shannon’s diversity index. In turn, the evenness index (E) was the highest in soils collected from the *P. sylvestris*, which indicated significantly lower diversity in these soils. The most abundant classes of bacteria in all samples were Actinobacteria, Acidobacteria_Gp1, and Alphaproteobacteria. The classes Bacilli, Thermoleophilia, Betaproteobacteria, and Subdivision3 were dominant in the *B. pendula* bulk soil. Streptosporangiales was the most significantly enriched order in the *B. pendula* soil compared with the *A. glutinosa* and *P. sylvestris*. There was a significantly higher mean proportion of aerobic nitrite oxidation, nitrate reduction, sulphate respiration, and sulfur compound respiration in the bulk soil of peri-root zone of *A. glutinosa*. Our research confirms that the evaluation of soil biodiversity and metabolic potential of bacteria can be of great assistance in a quality and health control tool in the soils of forested areas and in the forest production. Identification of bacteria that promote plant growth and have a high biotechnological potential can be assume a substantial improvement in the ecosystem and use of the forest land.

## 1. Introduction

Forests are natural ecosystems with multi-layered plant communities, dominated by trees. Because forests have many functions—including production, ecological, and social—it is important to preserve their species diversity and good health of stands for future generations [1,2]. Soil is one of the basic elements of forest habitats that guarantees the proper growth and development of an ecosystem. The use of forest soil properties is the basis for sustainable and proper forest management [3].

One of the key elements of forest soil is microorganisms. Bacteria and fungi are an integral part of the soil environment and perform a number of positive functions in it. They affect the functioning of ecosystems, plant health, and soil structure and productivity [4,5]. In forest ecosystems, edaphic conditions, plants, and soil microorganisms are closely related [6,7]. These relationships are determined by the physical, chemical, and biological properties of the soil [8,9,10]. Moreover, the weather conditions in a given year and season have a significant impact on changes in the biological activity of soils. The properties of soils and their biochemical processes are regulated by soil microorganisms [11,12].

A forest habitat is defined as a set of relatively persistent climatic, topographic, water and soil factors creating conditions for forest life [3]. Undergrowth vegetation in forests deformed by humans often does not reflect the possibilities of habitats, but only the possibilities of surface soil levels, not taking into account their deeper layers, which are accessible to tree roots [13,14,15]. The relationships between soils and vegetation in forests have been the subject of many studies [16,17,18,19]. They mainly concern the search for relationships between soil types, subtypes, and types of natural vegetation, with much less focus on soil biology and the microbiological changes that occur there [20,21]. Moreover, significantly little research has been conducted in semi-deciduous forests. This type of forest accounts for a large percentage of forests in Poland. In our research, we chose the three most common tree species from a semi-deciduous forest: *Alnus glutinosa*, *Betula pendula*, and *Pinus sylvestris*.

Semi-deciduous forest soils are the result of many years of relationships with trees [3]. Organic matter systematically reaches the soil, and the specificity and rate of decomposition of organic matter depend on the tree species. Decomposition products, also differentiated depending on the nature of the stand, affect the soil in a natural way when they are introduced into the soil and participate in the formation of characteristic elements of its morphology. Organic matter and its decomposition products cause soil acidification, leaching of elements, supply of nutrients, and modification of sorption properties [22,23]. In turn, the effect of soil on vegetation is most notable in the characteristic species composition of the natural forest complex related to the properties of a given part of the soil [3,24]. In the almost common view of foresters, soil is the least changed, most stable indicator complex and could be used to diagnose a habitat in any case, even where there is no tree stand and typically developed forest floor vegetation [24,25].

The use of indicators of soil biological activity to assess its fertility and productivity is well known. The diversity of soil microorganisms could be considered for an overall assessment of the bacterial composition and comparison among samples, or it could be used to identify the composition of the soil microbial community [26]. If the goal is to understand the function of the microorganisms and possible changes in it, it is sufficient to use known fingerprint methods, such as Biolog EcoPlates. Detailed knowledge of the composition of the soil microorganism composition requires a different analytical approach compared with the DNA extracted from soil [16].

Metagenomic is a method of analyzing the structure of microorganisms inhabiting a given microbiome (e.g., forest soil) [27,28]. By sequencing variable fragments in genomes, one can understand the genetic diversity, population composition, and ecological impact of microorganisms in the studied environment [29,30]. Information obtained from sequencing, interpreted in terms of the bacterial composition and maintaining its homeostasis, may constitute valuable knowledge regarding environmental protection of the forest ecosystem [31,32]. The state of knowledge regarding research on the biological activity of soils in individual forest habitats is insufficient. Currently, in forest management, these indicators are used to a negligible extent [33,34]. Little is known in particular about the composition of microorganisms inhabiting the peri-root zone of trees and about their functions.

The aim of this study was to determine the structural and functional diversity of composition of microbial population of bacteria in the bulk soil from peri-root zone of three selected tree species: the black alder (*Alnus glutinosa*), the white birch (*Betula pendula*), and the Scots pine (*Pinus sylvestris*). The hypothesis is that each soil taken in the proximity of the root system of tree species has its own unique bacterial composition and related metabolic functions.

## 2. Results

### 2.1. Field Site and Soil Chemical Analyses

Soil samples were collected in August 2019 and 2020 from a semi-deciduous forest located in an area belonging to the Agricultural Experimental Station IUNG-PIB in Osiny, Poland (51°27′57.3″ N 22°02′14.3″ E). The dominant tree species in the study area are *P. sylvestris*, *B. pendula*, and *A. glutinosa*. Samples were taken in the proximity of the root system tree root layers (0–20 cm) according to the Polish Standard [35]. The peri-root zone soil covers of the study area are mainly formed from postglacial sands. According to the World Reference Base soil classification system, the dominant soil types in this area are *Arenosols* and *Brunic Arenosols.*

Analysis of the particle size distribution showed that the sand fraction dominated all soil samples, with >90% in all samples (Table 1). The investigated soils had a very low clay fraction (<1%) (Table 1). The soils had a very low organic carbon content, with values ranging from 0.640% to 1.830% depending on the tree species (Table 1) [36]. The bulk soil from *P. sylvestris* had the lowest organic carbon content, while the soil from *B. pendula* had the highest content. The sandy forest soils were very poor in the macroelements phosphorus, potassium, and magnesium (Table 1). However, there was some difference in the phosphorus content depending on the stand type. There was a higher phosphorus content in *B. pendula* stands (10.192 mg P_2_O_5_ ∙ 100 mg^−1^ of soil) and *A. glutinosa* stands (6.170 mg P_2_O_5_ ∙ 100 mg^−1^ of soil) compared with *P. sylvestris* stands (1.115 mg P_2_O_5_ ∙100 mg^−1^ of soil) (Table 1). For potassium, *B. pendula* stands had the highest content (6.354 mg P_2_O_5_ ∙ 100 mg^−1^ of soil) compared with *P. sylvestris* stands (1.280 mg P_2_O_5_ ∙ 100 mg^−1^ of soil) and *A. glutinosa* stands (1.014 mg P_2_O_5_ ∙ 100 mg^−1^ of soil) (Table 1). In the case of magnesium, *B. pendula* stands had the highest content and *A. glutinosa* stands had the lowest (Table 1). The same relationship was noted for total nitrogen content: *B. pendula* stands had more of this component compared with the other tree species (Table 1). All investigated soil samples exhibited a clear acids pH. However, the lowest pH values were recorded under the pine stand (Table 1).

### 2.2. Enzymatic Activities 

The soils collected from the peri-root zone of black alder, silver birch, and Scots pine had high enzymatic activities. Determination of DHA, AcP, and AlP in the soil samples provided us with a large amount of information about the biological characteristic of the soil. Table 2 shows the results of the dehydrogenase, acid, and alkaline phosphatase activities in soils collected from the bulk soils in 2019–2020. The years 2019 and 2020 were very similar in terms of weather conditions (temperature and rainfall). Nevertheless, the weather conditions were of great importance in determining the enzymatic activity of soils. The highest dehydrogenase activity was found in 2019 in soils collected from the *A. glutinosa* stands (26.270–51.504 μg formazan/g dry matter (d.m.) of soil/24 h). In 2020 also, the highest dehydrogenase activity was also found in the *A. glutinosa* stands (19.952–25.067 μg formazan/g d.m. of soil/24 h). The bulk soil collected from *P. sylvestris* stands had the lowest dehydrogenase activity in both 2019 and 2020. Similar results were obtained in the case of phosphatase activity. The bulk soils collected from the *A. glutinosa* stands in both 2019 and 2020 had the highest acid phosphatase activity. The *P. sylvestris* stands had significantly lower acid phosphatase activity (13.778–18.471 μg p-nitrophenol/g d.m. of soil/h). The phosphatase and dehydrogenase activities in the *B. pendula* bulk soil were between the *A. glutinosa* and *P. sylvestris* (Table 2).

### 2.3. Metabolic Profiles of Bacterial Community Based on Biolog EcoPlates

Metabolic profiles of bacterial community based on Biolog EcoPlates provide useful information about environmental changes in the peri-root zone of trees. The utilization percent of use of selected groups of substrates obtained in the Biolog EcoPlates incubated for 120 h is presented in Figure 1. 

The highest utilization percent of amines and amides (Figure 1A) and amino acids (Figure 1B) was observed in the bulk soil from *A. glutinosa* stands. In turn, in the bulk soil from B. pendula stands, the best utilized compounds were carboxylic and acetic acids (Figure 1C). The best utilized compounds in the bulk soil of *P. sylvestris* were carbohydrates (Figure 1D). Compounds belonging to the group of polymers were most effectively used by microorganisms in the bulk soil from *A. glutinosa* stands (Figure 1E).

Figure 2 shows heat maps based on the analysis of 31 carbon sources after 120 h of incubation of the Biolog EcoPlates. The soils collected from the peri-root zone of *A. glutinosa* were characterized as the most actively utilized such as compounds 2-hydroxy benzoic acids, i-erythritol, D-xylose, putrescine, beta-methyl-D-glucoside, alpha-cyclodextrin, glucose-1-phosphate, alpha-D-lactose, L-serine, and DL-alpha-glycerol-phosphate. These results were similar in 2019 and 2020. On the other hand, the slowest utilized compounds in bulk soil were D-mannitol, D-galacturonic acid, gamma-lactone, L-arginine, N-acetyl-D-glucosamine, and D-glucosaminenic acid (Figure 2). The soils collected from the peri-root zone of *P. sylvestris* stands were characterized as the lowest utilization of the 31 compounds. Determination of community-level physiological profiles using Biolog EcoPlates effectively distinguishes temporal changes in microbial communities, especially in terms of the year and place of sampling.

On the basis of the data obtained from the absorbance measurements during the utilization of 31 different carbon sources, we calculated the average well color development (AWCD) index (Figure 3). The highest AWCD values were obtained for samples collected from the bulk soil collected from the peri-root zone of *A. glutinosa* stands. The metabolic activity in these soils was high after 24 h of incubation of Biolog EcoPlates and remained at the highest level until 120 h of incubation. In the case of soils collected from the *B. pendula* and *P. sylvestris* stands, an increase in metabolic activity was observed only after 48 h. The soils collected from the bulk soil of *P. sylvestris* stands had the lowest metabolic activity.

Changes in diversity index values calculated on the basis of the absorbance values of Biolog EcoPlates after 120 h of incubation are presented in Table 3. For all selected indicators—Shannon’s general diversity index (*H*’), substrate richness (R), substrate evenness (E), and AWCD_590_—the same relationships were observed both in 2019 and 2020. The bulk soil collected from the peri-root zone of *A. glutinosa* and *B. pendula* stands had the highest diversity indicator values (Table 3). The soil collected from peri-root zone *P. sylvestris* had the lowest metabolic activity according to the above-mentioned indicators. In turn, the substrate evenness index (E) was highest in the *P. sylvestris* stands, which indicated a significantly lower metabolic biodiversity in these soils.

The correlation between samples, soil metabolic activity, and biodiversity indices were assessed by principal component analysis (PCA), which was performed separately for 2019 and 2020 to account for the variability in activity over time. A similar tendency was shown in the PCA analysis, regardless of the year of the study. In 2019, the first two principal components (PCs) accounted for 85.65% and 12.59%, respectively, of the total variance in the bulk soils samples from the *A. glutinosa*, *B. pendula*, and *P. sylvestris* stands (Figure 4A). In 2020, the first two PCs accounted for 75.48% and 12.65%, respectively, of the total variance (Figure 4B). The PCA plot of the first two PCs showed that the samples clustered by the tree species. All metabolic activity indices (Shannon, AWCD, richness, amines and amides, amino acids, carboxylic and acetic acids, carbohydrates, and polymers) were positively related to samples collected from the *A. glutinosa* and *B. pendula* stands. By contrast, samples collected from peri-root zone of *P. sylvestris* were negatively related to the soil metabolic activity and positively related to the evenness index (Figure 4).

### 2.4. 16S rRNA Sequencing

Biodiversity indices were calculated on the basis of the data obtained from 16S rRNA sequencing. Figure 5 shows the alpha diversity index values for each of the bulk soil samples collected in 2019 and 2020. The *A. glutinosa* stands had the highest diversity (species richness and Shannon diversity). The biodiversity indicators were approximately the same in the *B. pendula* and *P. sylvestris* stands. 

The most abundant phyla were Proteobacteria, Actinobacteria, and Acidobacteria, with a lower percentage of Bacteroidetes, Firmicutes, and Verrucomicrobia (Figure 6A).

Firmicutes was not found in soil collected from the peri-root zone of the *A. glutinosa* stands. On the other hand, Bacteroidetes and Verrucomicrobia were more abundant in the *B. pendula* and *P. sylvestris* stands. Rhodospirillales, Gammaproteobacteria, Mycobacteriales, Bacillales, and Rhizobiales were the most abundant orders (Figure 6B). Rhizobiales was most abundant in the *A. glutinosa* and *B. pendula* stands. The most abundant bacterial classes were Actinobacteria, Acidobacteria_Gp1, and Alphaproteobacteria (Figure 6C). The classes Bacilli, Thermoleophilia, Betaproteobacteria, and Subdivision3 were dominant in the *B. pendula* stands.

The dendrogram at the genus level was constructed by using the unweighted pair group method with arithmetic mean (UPGMA) algorithm (Figure 7). All soil samples, regardless of the sampling year, were grouped depending on the type of tree. The result may indicate a close relationship between the structure of the bacteria in soils collected from the peri-root zone and a tree species: *A. glutinosa*, *B. pendula*, and *P. sylvestris*.

The linear discriminant analysis effect size (LefSE) analysis revealed that samples collected in 2019 had more enriched bacterial taxa than samples collected in 2020. The phyla Firmicutes, Nitrospirae, and Verrucomicrobia and their members were significantly enriched in the *A. glutinosa* stands compared with other tree species in 2019. The Planc-tomycetes phylum and its member were significantly enriched in *B. pendula* samples. In the *P. sylvestris* stands, no phylum was significantly enriched, but the acidobacteria_Gp13 and Alphaproteobacteria classes were significantly enriched. Interestingly, the phylum Proteobacteria was significantly enriched considering all tree species, but the families from this phylum were differentially enriched: *Bradyrhizobiaceae, Hyphomicrobiaceae*, *Rey-anellaceae*, *Erythrobacteraceae*, and *Alcaligenaceae* in the soils collected from the peri-root zone of *A. glutinosa*, *Roseiarcaceae* in the *B. pendula* stands, and *Acetobacteraceae* and *Xanthomona-daceae* in the *P. sylvestris* stands.

Compared with 2019, the tree species of 2020 harbored fewer significantly different bacterial taxa. The number of significantly enriched bacterial taxa from 2020 was again highest from *A. glutinosa* samples, which included the phylum Nitrospirae and its member, class Acidobacteria_Gp6. Unlike 2019, the phylum Proteobacteria and its classes Alphaproteobacteria and Gammaproteobacteria were enriched in the *P. sylvestris* samples. The only order enriched in the *B. pendula* stands compared with *A. glutinosa* and *P. sylvestris* was Streptosporangiales (Figure 8). 

The three main groups were distinguished according to the tree species on the basis of the analysis NMDS (non-metric multidimensional scaling dissimilarity) (Figure 9). Strong positive correlations were found between the tree species and selected types of bacteria. Regardless of the year of sampling, each tree species had its own characteristic bacterial composition. Bulk soil of *Alnus glutinosa* was characterized by strong positive correlations with the following genera of bacteria: Gp2, Subdivision3_genera_incertae_sedis, *Rhodonobacter*, *Roseiarcus*, unclassified_050, unclassified_056, and unclassified_058. The following genera dominated in the *Betula pendula* stands: *Acidiferrimicrobium*, Gp1, *Acidibacter*, WPS-1_genera_incertae_sedis, unclassified_0083, unclassified_0087, and unclassified_0105. The following genera showed strong correlation with addition to the *Pinus silvestris* samples: *Mycobacterium*, *Silvibacterium*, *Chujaibacter*, *Conexibacter*, Gp13, Gp14, unclassified_049, unclassified_052, and unclassified_053.

It was also very important to determine the functions that these bacteria can perform in a given soil. According to FAPROTAX, 28.6% of bacterial genera (178 of 622) were assigned to at least one function group (38 groups in total; Figure 10). The significantly higher mean proportion of aerobic nitrite oxidation, nitrate reduction, sulfate respiration, and sulfur compound respiration in *Alnus glutinosa* samples were found in comparison to other tree categories from the year 2019. However, from 2020, only fermentation was found to be significantly higher in samples of *Alnus glutinosa.* Unlike the bulk soil of *Alnus glutinosa*, we did not find any significant differences in the mean proportion of bacterial putative function in *Betula pendula* (Bp) and *Pinus silvestris* (Ps) compared to all other samples from year 2019. Interestingly, aerobic chemoheterotrophy and chemoheterotrophy had a higher mean proportion of bulk soil of *Betula pendula* in the year 2020, which was not evident in the year 2019. The nitrogen fixation had a significantly higher mean proportion in 2020 in samples of *Pinus silvestris* (Figure 10).

## 3. Discussion

Forests constitute one of the largest and most important ecosystems on Earth, covering over 40 million km^2^ and accounting for 30% of the total land area in the world [1,24]. The ecology of forest soils is an important area of research due to the role of forests as carbon sinks. Hence, a significant amount of information has accumulated regarding their ecology, especially for temperate climates and boreal forests [20]. Although most of the research has focused on fungi, soil bacteria also play an important role in this environment. In forest bulk soils, bacteria inhabit many habitats with specific properties, including soil, rhizosphere, litter, and deadwood habitats where their communities are shaped by nutrient availability and biotic interactions [20]. In the peri-root zone of forest trees, bacteria interact with plant roots and mycorrhizal fungi as commensalisms or mycorrhiza helpers. Understanding the ecology of bacteria in forest soils has advanced dramatically in recent years, but it is still incomplete [37].

Trees are associated with their own microbiome and mycobiome [10,17,21]. The peri-root zone of a tree comprises numerous bacteria that potentially promote plant growth, and it represents a valuable resource for sustainable and ecological agriculture [38,39]. Soil microorganisms play a major role in soil quality and functioning, largely determining soil structure and nutrient cycling, and ultimately impacting plant performance through nutrient mobilization, especially P, root growth, and plant health [40]. Previous studies have highlighted the importance of diverse belowground systems to maintain the stability and productivity of ecosystems and their multifunctionality [41]. 

Community-level physiological profiles using Biolog MicroPlates were originally described in 1991 by J. Garland and A. Mills [42]. This analysis effectively distinguishes spatial and temporal changes in microbial communities. In applied ecological research, EcoPlates have been used in many studies as both an assay of the stability of a normal population and to detect and assess changes following the onset of an environmental variable [43,44,45,46]. 

We found that the peri-root zone soils of each of the tree species differed significantly in terms of metabolic profile. The highest metabolic diversity was confirmed for the *A. glutinosa* and *B. pendula* samples. On the other hand, the *P. sylvestris* stands had the lowest metabolic activity. Other authors have confirmed the statistical dependence of soil metabolic activity on the plant species and the physicochemical and biological properties of soil [45]. The bulk soil is a special microenvironment with very active of nutrient circulation and microbial community formation [41,44].

Si et al. [47] examined the soils of eight common deciduous fruit trees in northern China and showed that the type of vegetation was the main factor influencing changes in biodiversity of bulk soils, calculated by soil physicochemical properties, enzyme activities, and the community-level physiology. Our research also confirmed that the bacterial composition of bulk soil from peri-root zone of selected tree is crucial. This is confirmed both by the results of the research on the biological activity of these soils as well as by the study of the metabolic profile and the structure of the microbiome. 

As reported by other authors, many biotic and abiotic factors influence the biodiversity of forest soils [7,10,12,15]. Thiem et al. (2018) [48] showed that depending on the degree of salinity in the soil collected from *A. glutinosa*, the dominant genera were *Rhodanobacter*, *Granulicella*, and *Sphingomonas* (in saline sites), and *Bradyrhizobium* and *Rhizobium* (in non-saline sites). These genera also dominated in our study of the *A. glutinosa* samples (an actinorhizal plant). Many authors have confirmed that the forest soil microbiome may provide significant advantages to the plant [21,32,33]. These changes can be influenced by many factors, including moisture, temperature, soil pH, oxygen, salinity, soil contamination, drought, and trace elements. In addition, tree species can also be of great importance to the changes taking place in the bacterial composition of soils. This was confirmed by our research because, in the same type of soil, the microbiome differed significantly depending on the tree species. Tree roots attract their associated microbial partners from the local soil community. Accordingly, tree-root-associated microbial communities are shaped by both the host tree and local environmental variables [15,30]. The abundance of elements such as carbon, nitrogen, phosphorus, potassium, and magnesium in soils also affects the composition and functions of microorganisms in the bulk soil of trees [14]. Soil pH correlates with microbiome composition, with the microbial assemblages changing in more acidic soils [8,34,49].

Peng et al. [49] showed that factors such as soil pH, invertase activity, and nitrogen content had a significant impact on soil microbial community. Invertase activity was positively correlated with plant-beneficial microbes such as *Mortierella*, *Geomyces*, *Lysobacter*, and *Chaetomium*. In our research, there was high dehydrogenase and acid phosphatase activity in the *A. glutinosa* and *B. pendula* samples, which were dominated by Actinobacteria, Acidobacteria_Gp1, Alphaproteobacteria, Bacilli, Thermoleophilia, and Betaproteobacteria.

In our previous research, we conducted an assessment of soil biological activity, including enzymatic activity, in eight different soil types. These soils differed in physico-chemical and biological properties, including a pH range of 4–7.5 [50]. The soil profiles were collected with preserving the natural soil horizons. The microplot experiment was carried out on different soil types: Brunic Arenosol (Dystric I), Rendzic Leptosol, Fluvic Cambisol, Haplic Cambisol (Eutric), Gleyic Phaeozem, Brunic Arenosol (Dystric II), Haplic Cambisol (Eutric II), and Haplic Cambisol (Dystric). The enzymatic activity was strictly dependent on the type of soil and soil pH. The highest dehydrogenase and alkaline phosphatase activities were found in Gleyic Phaeozem (pH 7.4) and the lowest in Brunic Arenosol (Dystric I and II; pH 4, 4.5). The highest acid phosphatase was found in Brunic Arenosol (pH 4, 4.5). Other authors also confirmed this dependence in their research [51,52,53].

Lemanowicz and Bartkowiak [53] assessed the enymatic activity in salt-affected soils. The pH range of soils was 4.45–7.40. Alkaline phosphatase activity was much lower in soils with pH 4–5. It was confirmed that although oxygen and other electron acceptors can be utilized by dehydrogenases, most of the enzyme is produced by anaerobic microorganisms. The activity of acid phosphatase is related to the presence of soil microorganisms inhabiting a given soil [50,52]. On the other hand, the activity of alkaline phosphatase is related to plant enzymes. In acidic soils, a significant increase in the activity of acid phosphatase is observed, which is correlated with the activity of soil microorganisms. The activity of alkaline phosphatase in acidic soils is at a lower level, as it is mainly related to the activity of microorganisms preferring the neutral pH of the soil and is related to plants. In our research, we evaluated acid forest bulk soils that are not closely related to tree roots, and hence we showed a much lower activity of alkaline phosphatase compared to acid phosphatase activity.

Researchers suggest that plant types and species harbor partially different microbiomes [26,29]. These changes are mainly observed in the peri-root zone and rhizosphere of plant roots, which we have confirmed in this study. For example, grapevine rootstock genotypes in mature vineyard were associated with different microbiomes [41]. The authors proved during two years of research using 16S rRNA sequencing that bacterial microbiota across five grapevine rootstock genotypes cultivated in the same soil depended on the plant genotype [32,34,54].

Other studies have confirmed that the genus *Bradyrhizobium* is the dominant genus and symbiote of Australian acacia [14]. *Acacia dealbata* enriches bulk soils with potentially beneficial microbial taxa and members of the genus *Bradyrhizobium* may play an integral role in this process. In our study, the taxa Rhizobiales was most abundant in the *A. glutinosa* and *B. pendula* samples. As reported by other authors, soil samples collected from scots pine contained more Planctomycetia, Actinobacteria, Opitutae, and Deltaproteobacteria, whereas Sphingobacteriia, Gammaproteobacteria, Armatimonadia, and Betaproteobacteria were more common in the *P. sylvestris* samples [55]. Other research confirmed that the *P. sylvestris* endorhizosphere is enriched in the genera *Pseudomonas*, *Burkholderia*, and *Bacillus* [56]. This dependence was also confirmed in our research. 

The health condition of a tree may also impact the biodiversity of the bacteria in soils. Pinho et al. [57] found significant changes in the biological activity and microbiome composition across oak health conditions. The authors evaluated three disease stages of plants (low, mid, and severe) and confirmed significant changes in the composition of the root bacterial microbiome depending on the stage of the disease. Members of the phyla Nitrospirae, Chloracidobacteria, and Acidobacteria subdivision 6 were consistently more abundant in healthy trees. We confirmed this finding in our research in which we also examined healthy trees. Moreover, Acidobacteria are active members in the peri-root zone of tree, a finding that confirms previous research.

Better understanding the changes in the composition of microbial population of bacteria in bulk soil from the peri-root zone of forest trees and their metabolic processes is key to providing in-depth knowledge to develop more efficient, handy, sustainable, and reliable forest management strategies. It is also crucial to follow these changes in a given climatic zone and a specific type of forest. This study had provided preliminary evidence of changes in the bulk soil of three tree species and confirms the uniqueness of the composition of bacteria depending on the species. In our future research, we will also explore the role of fungi in this process. The root zone of plants is a wealth of diversity not only of bacteria but also fungi.

## 4. Materials and Methods 

### 4.1. Sample Collection

Soil samples were collected from the semi-deciduous forest area belonging to Agricultural Experimental Station in Osiny, Puławy, in the southeast of Poland (51°27′57.3″ N 22°02′14″ E) (Figure 11). Semi-deciduous forest is characteristic of the habitat type and is found in mainly the lowlands. In Poland, this type of forest covers about 10.5% of the forest habitat area. This type of forest is found on various soil formations, but most commonly on sandy soils such as Brunic Arenosols and Arenosols. In mixed forests, there are mainly clay sands and sandy loams. In a semi-deciduous forest habitat, the main tree species are pine, birch, alder, oak, beech, spruce, fir, and larch. We chose the three most common tree species for our research: *Alnus glutinosa* (Ag), *Betula pendula* (Bp), and *Pinus sylvestris* (Ps). Temperate forests are characterized by temperature ranges between 20 and 30 °C, with hot summers and cold winters; the average annual temperature for the area is 6.5 °C, and average annual precipitation is 726 mm. 

Three plants were selected for each tree species. Figure 12A shows the characteristic habitat for each tree species. The bulk soils from the peri-root zone were collected separately for each tree in summer (August) 2019 and 2020. First, the top layer of the leaf litter was removed. Then, a pooled sample of the rhizosphere soil within the tree roots was collected (Figure 12B). The soil samples were collected from the 0–20 cm layer in three replicates and sieved through a 2 mm sieve and stored in a refrigerator (4 °C) until analysis [35].

### 4.2. Soil Physico-Chemical Analysis

Several physicochemical properties were measured in the air-dried samples: texture using a Mastersizer 2000 laser diffraction particle size analyzer (Malvern Instruments, Worcestershire, UK); soil organic carbon (C) and total nitrogen (N) by combustion with a vario Macro 8 cube CN analyzer (Elementar, Langenselbold, Germany). The level of the exchangeable cation Mg was determined by using an AAnalyst 800 atomic absorption spectrometer (AAS) (Perkin Elmer, Waltham, MA, USA). The available P_2_O_5_ was determined by the Egner–Riehm colorimetric method, using extraction with 0.02 M calcium lactate in 0.01 M HCl, followed by colorimetric measurement in a Lambda 45 Spectrometer (Perkin Elmer, Waltham, MA, USA), on the basis of the reaction with ammonium molybdate and available K_2_O by the Egner–Riehm method, with K measurement using an AAnalyst 800 AAS. The ammonium and nitrate solubility were determined by flow spectrometry after extraction with 1M K_2_SO_4_ by using a QuAAtro39 analyzer (Seal Analytical, Norderstedt, Germany).

### 4.3. Enzymatic Activities 

Enzymatic activity was determined spectrophotometrically: soil dehydrogenase activity using the 2,3,5-triphenyltetrazolium chloride (TTC) method [58] and phosphatase activity by the *p*-nitrophenyl phosphate (*p*-NPP) method [59]. Soil dehydrogenase activity (DHA) was determined in 6 g of soil by colorimetric measurement of the reduction of TTC solution to triphenylformazan (TPF) after incubation at 37 °C for 24 h. All microbial analyses were carried out as three replicates [56]. The acid phosphatase (AcP) and alkaline phosphatase (AlP) activities were analyzed by using 1 g of soil incubated for 1 h (37 °C) with *p*-nitrophenyl phosphate at their optimum pH of 6.5 and 11, respectively, using the spectrophotometric method. 

### 4.4. Biolog EcoPlates 

The Biolog EcoPlate system (Biolog Inc. Hayward, CA, USA) was used to determine the metabolic potential of microorganisms in the bulk soil from the peri-root zone of trees. The analysis was conducted by following the manufacturer’s instructions. Metabolic profiling of all soil samples was carried out by using the Biolog EcoPlates, which contain 31 different carbon sources divided into five compound groups: amines and amides, amino acids, carboxylic and acetic acids, carbohydrates, and polymers [38]. One gram of soil was suspended in 99 mL of sterile water, shaken for 20 min at 20 °C, and incubated at 4 °C for 30 min [42,60,61]. Next, each well of a Biolog EcoPlate was inoculated with 120 μL of the prepared suspension and incubated at 25 °C. Absorbance at 590 nm was measured on a Biolog Microstation after 24, 48, 72, 96, and 120 h of incubation. The analysis was carried out as three replicates.

### 4.5. DNA Extraction, Amplification, and Next-Generation Sequencing

Fresh soil samples were weighed, and 300–350 mg of fresh soil was collected into 1.5 mL tubes for extracting DNA with the FastDNA™ SPIN Kit for Soil (MP Biomedical, Hayward, CA, USA), according to the manufacturer’s instructions. DNA purity and concentration were measured with a NanoDrop 1000 Spectrophotometer (Thermo Scientific, USA). DNA was diluted with sterile MilliQ water to 10 ng µL^−1^. Soil samples collected in 2019 were sequenced at Genomed S.A (Warsaw, Poland) with 2 × 250 base pair (bp) paired-end technology using the Illumina MiSeq system. Soil samples collected in 2020 were sequenced at Department of Plant Breeding and Biotechnology IUNG-PIB in Puławy, Poland with the same methodology with 2 × 300 base pair (bp) paired-end technology using the Illumina MiSeq system.

The two-step PCR amplification was utilized. In the first-step PCR, we amplified hypervariable V3–V4 region of the 16S rRNA gene with using 341F and 785R primers [62]. The reaction was carried out in 50 µL volume containing 12.5 ng template DNA, 0.1µM of each primer, and KAPA HiFi HotStart ReadyMix (2X) (Roche, Kapa Biosystems, Cape Town, South Africa). The thermocycling conditions were initial denaturing at 95 °C for 3 min, followed by 25 cycles of denaturation at 95 °C for 30 s, 55 °C for 30 s, 72 °C for 30 s, and final extension at 72 °C for 5 min. In second step, we attached dual indices and Illumina sequencing adapters using the Nextera XT Index Kit (Illumina, San Diego, CA, USA), according to the manufacturer’s instructions. The clean-up procedure using AMPure XP beads (Beckman Coulter, Indianapolis, IN, USA) was applied after every PCR step. Final library size validation was performed using the 2100 Bioanalyzer, 1000 DNA chip (Agilent Technologies, Santa Clara, CA, USA). Libraries were quantified using QuantiFluor^®^ dsDNA System on Quantus Fluorometer (Promega, Madison, WI, USA).

Sequencing of pooled libraries was carried out using Illumina 2 × 300 bp kit on a full run of the Miseq instrument (Illumina, San Diego, CA, USA). PhiX control at 5% was spiked into the run to add base diversity.

The 16S rRNA amplicon sequencing data generated in this study was deposited in the NCBI Sequence Read Archive (SRA) under the BioProject number PRJNA779918 (Table 4).

### 4.6. Bioinformatic and Statistical Analysis

Demultiplexed fastq files were processed using the DADA2 (1.14) package [63] in R software (3.6.0) [64]. On the basis of the quality plots, we trimmed the last 20 and 70 bases off forward and reverse reads, respectively. Primers sequences were removed from all reads. Filter parameters were as follows: maxN = 0, maxEE for both reads = 2, truncQ = 2. The error rates were estimated by learnErrors using 1 million reads, and exact sequence variants were resolved using dada. Next removeBimeraDenovo was used to remove chimeric sequences. After the filtration steps, an average of 132,532 reads were left for further analysis. Taxonomy was assigned against the latest version of the modified RDP v18 database [65] (http://www2.decipher.codes/Classification/TrainingSets/RDP_v18-mod_July2020.RData, accessed on 19 July 2021) using IDTAXA [66] on the sequences table resulting from the DADA2 workflow described above. The results were converted and imported into the phyloseq (1.22.3) package [67]. Sequences belonging to the chloroplast or mitochondrial DNA were removed. Subsequently, for further analysis, the total number of reads for the individual taxa were converted to a percentage, assuming the sum of all taxa in the individual samples as 100%.

The microbiome package in R 3.5.3 was used to calculate bacterial alpha diversity indices, which included species richness (Chao1), diversity (Shannon), and evenness (Simpson) [28]. Each identified ASVs were also annotated to predictive functions potential of bacteria using the functional annotation of prokaryotic taxa (FAPROTAX) pipeline [68]. FAPROTAX is a manually constructed database that maps prokaryotic taxa (e.g., genera or species) to putative functions on the basis of the literature of cultured representatives. The STAMP bioinformatic package was used to identify the significant difference in mean proportion of bacterial putative functions between individual tree type with all other samples from each year [31]. 

Differences in bacterial family abundance among different tree types of each years were tested using the nonparametric Kruskal–Wallis sum-rank test and the unpaired Wilcoxon test. These tests were followed with linear discriminant analysis (LDA) to estimate the effect size of taxonomical covariates driving the group difference procedure implemented in LEfSe (linear discriminant analysis effect size) [32]. This tool allows the analysis of microbial community data at any clade (from phylum to species level); however, the analysis of the large number of ASVs detected in this study would be computationally too complex, and therefore statistical analysis was performed only from the domain to the family level, and only significant ASVs were used to plot the cladograms. For LEfSe analysis, statistically significant alpha values for the factorial Kruskal–Wallis test among classes and for the pairwise Wilcoxon test between subclasses were set to 0.05. The threshold on logarithmic LDA score for discriminative features was set to 3 to identify the bacterial families with a statistically significant difference. The all-against-all strategy was used for LEfSe. 

The main statistical analyses for biological activity were performed using STATISTICA 10 (Stat. Soft. Inc., NY, USA). Collected data were subjected to analysis of variance (ANOVA) for comparison of means. Significant differences were calculated according to Tukey’s post hoc test at a *p* < 0.05 significance level. The cluster analyses were performed on standardized data from the average absorbance values at 120 h (Biolog EcoPlate). The results were also submitted to principal component analysis (PCA).

## 5. Conclusions

Our preliminary research has shown that the soil collected from the rhizosphere of selected tree species—black alder, silver birch, and Scots pine—had a unique bacterial microbiome. This microbiome was closely dependent on the tree species and the physicochemical properties of the soil. Soil biological activity and metabolic processes carried out by microorganisms also depended on soil properties and plant species. Understanding the diversity of bacteria in the rhizosphere and the interactions between microbiota and trees will facilitate the development of future strategies for forest protection and finding the appropriate indicators for reliable assessment of forest soils.

## Figures and Tables

**Figure 1 ijms-23-02633-f001:**
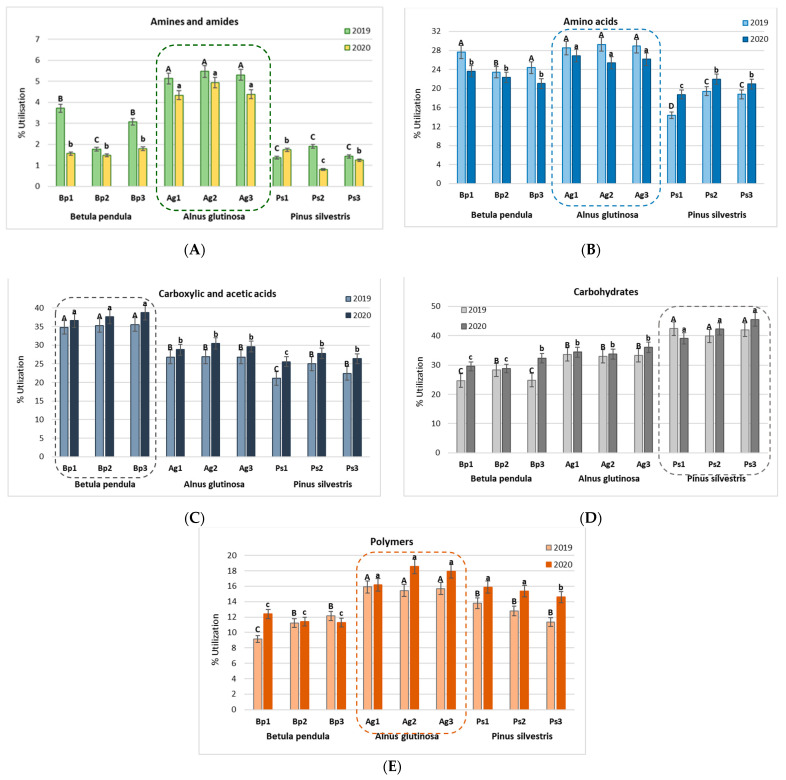
Utilization percent of selected groups of substrates obtained in the Biolog EcoPlates incubated for 120 h (*n* = 3): (**A**) amines and amides, (**B**) amino acids, (**C**) carboxylic and acetic acids, (**D**) carbohydrates, and (**E**) polymers. Treatment means separated by different letters (A, B, C, D for 2019 and a, b, c for 2020) are significantly different (Tukey’s mean separation test, *p* < 0.05); *n* = 3.

**Figure 2 ijms-23-02633-f002:**
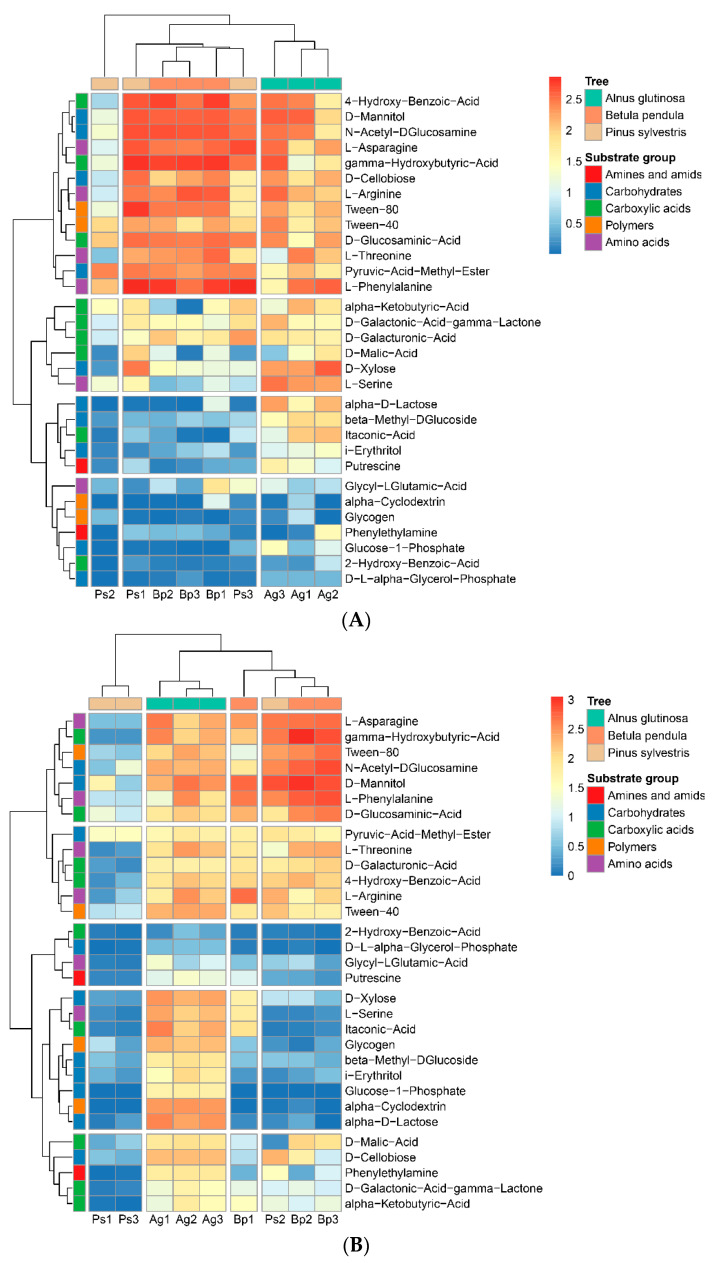
Heatmap based on the analysis of 31 carbon sources after 120 h of incubation of the Biolog EcoPlates. (**A**) The data obtained from samples collected in 2019. (**B**). The data obtained from samples collected in 2020. The results illustrate the difference in microbial communities in each sample according to the substrate utilization. The data were standardized (*n* = 3).

**Figure 3 ijms-23-02633-f003:**
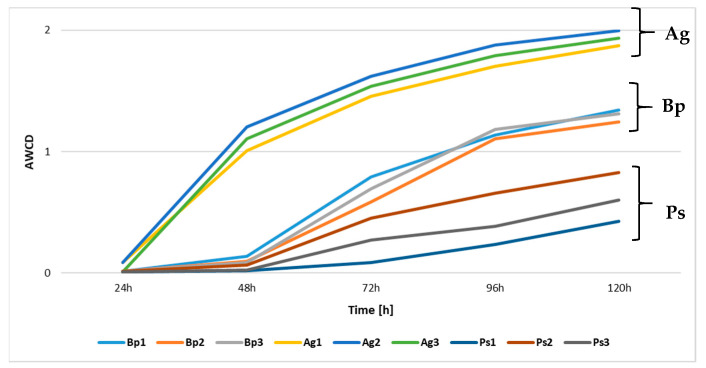
AWCD index values, two-year mean (*n* = 9), wavelength 590 nm.

**Figure 4 ijms-23-02633-f004:**
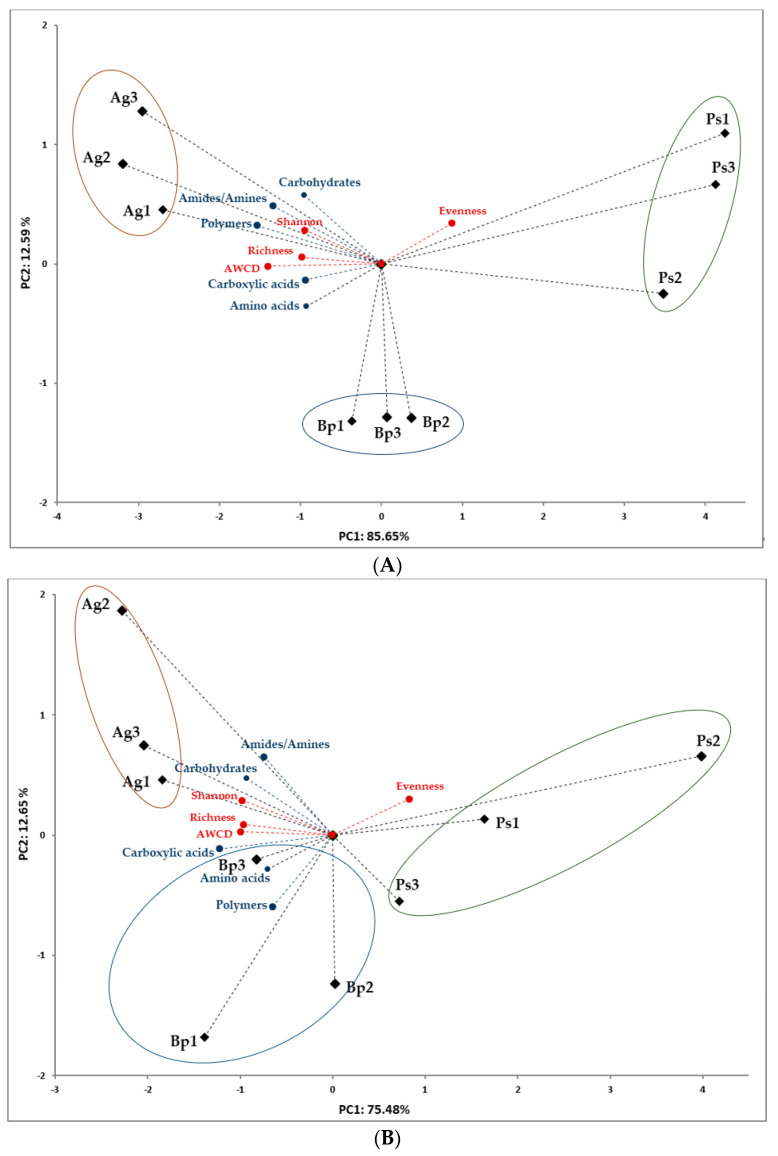
Principal component analysis (PCA). (**A**) Results obtained in 2019. (**B**) Results obtained in 2020.

**Figure 5 ijms-23-02633-f005:**
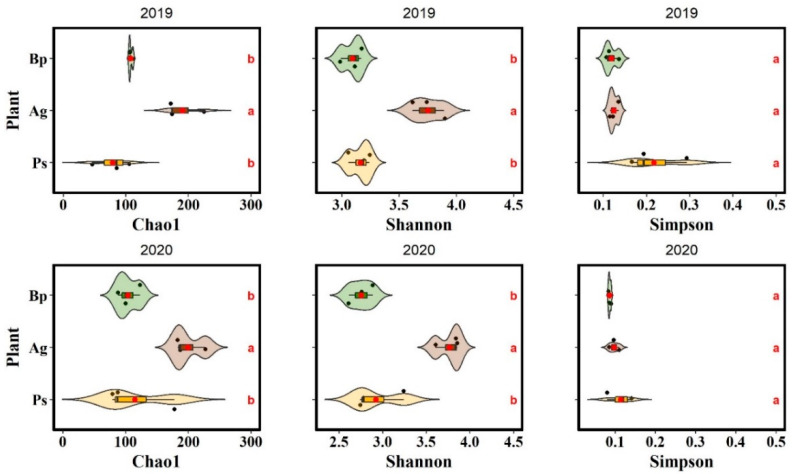
Alpha diversity indices: species richness (Chao1), diversity (Shannon), and evenness (Simpson).

**Figure 6 ijms-23-02633-f006:**
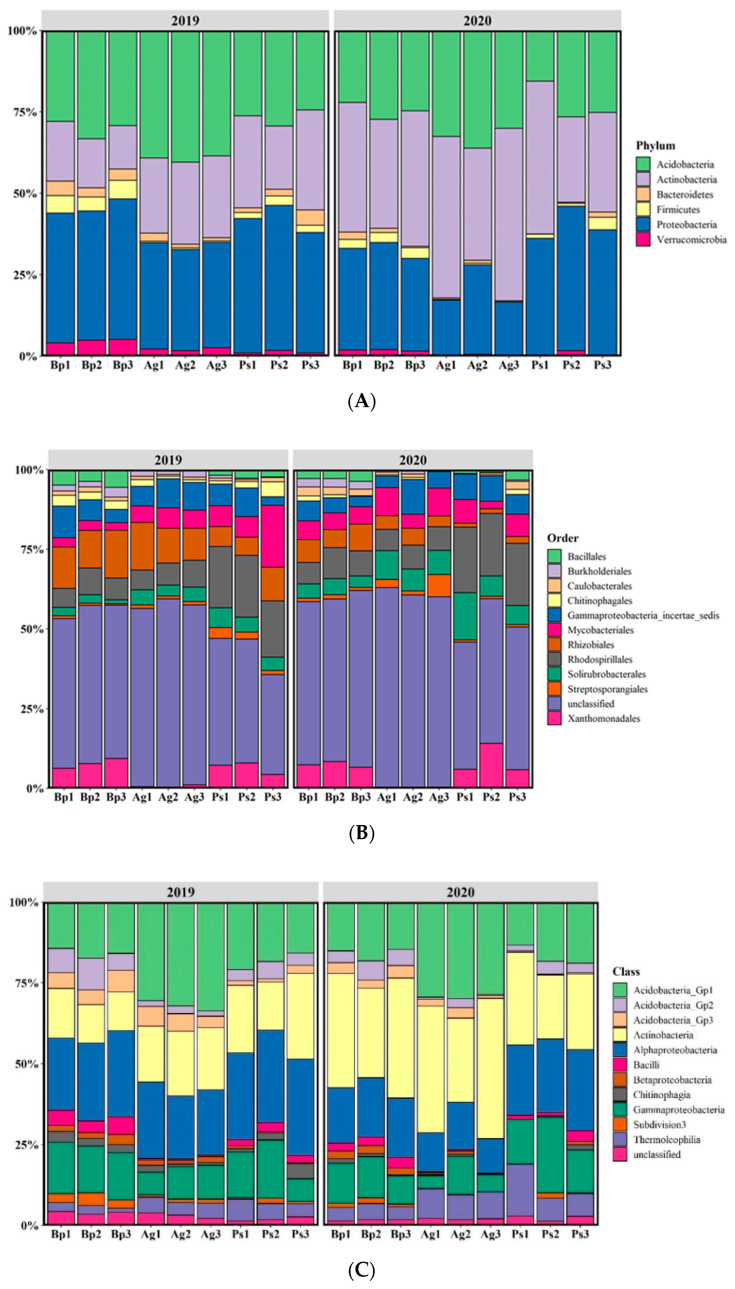
Relative proportion of dominant bacterial taxa in the bulk soils (percentage of sequences) according to next-generation sequencing (16S rRNA) by sampling site. (**A**) At the phylum level; (**B**) at the order level; (**C**) at the class level.

**Figure 7 ijms-23-02633-f007:**
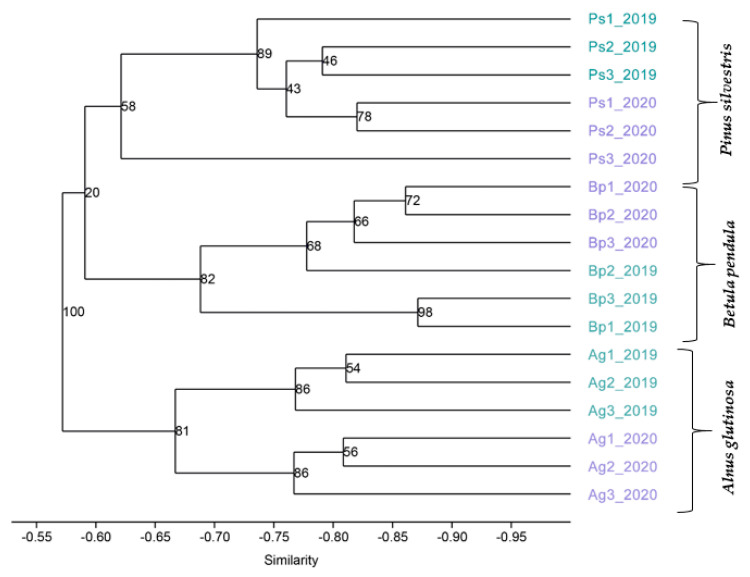
The UPGMA algorithm; the dendrogram with the pairwise similarity matrix, Bray Curtis dissimilarity.

**Figure 8 ijms-23-02633-f008:**
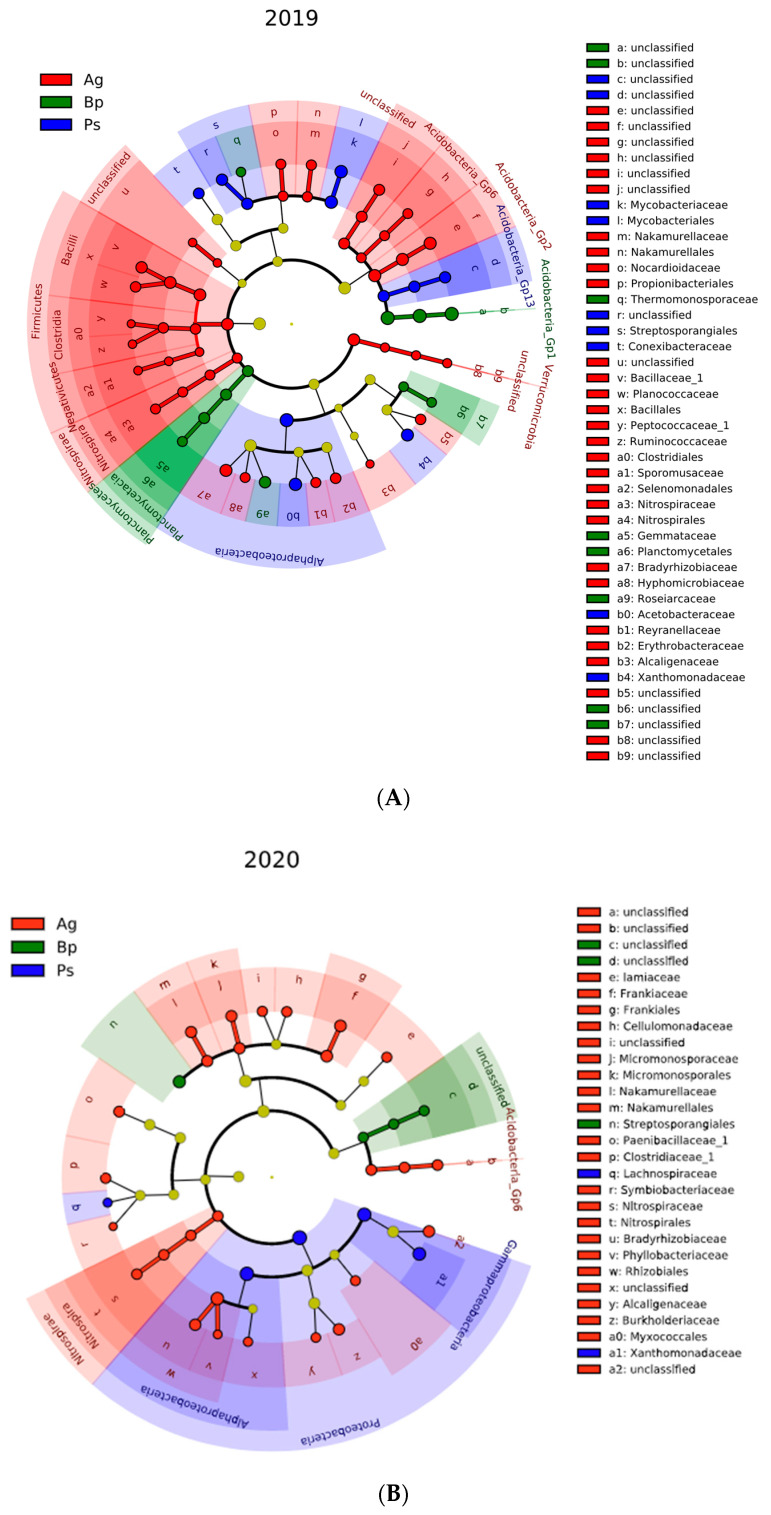
The LEfSe analysis representing differentially abundance families between groups: Alnus glutinosa (Ag), Betula pendula (Bp), and Pinus silvestris (Ps). (**A**) Summary for 2019; (**B**) summary for 2020.

**Figure 9 ijms-23-02633-f009:**
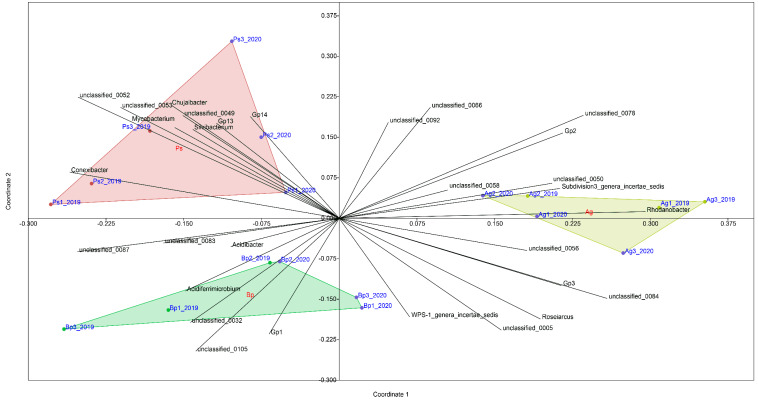
NMDS—non-metric multidimensional scaling dissimilarity measure among samples collected in 2019 and 2020.

**Figure 10 ijms-23-02633-f010:**
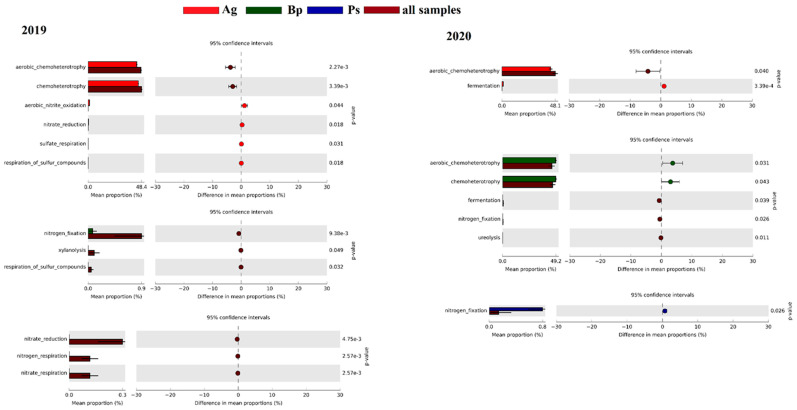
Significant difference in mean proportion of bacterial putative functions between individual tree type with the rest of the tree types from different years. Distinct colors represent different tree types: red, Ag; green, Bp; blue, Ps; brown, all samples.

**Figure 11 ijms-23-02633-f011:**
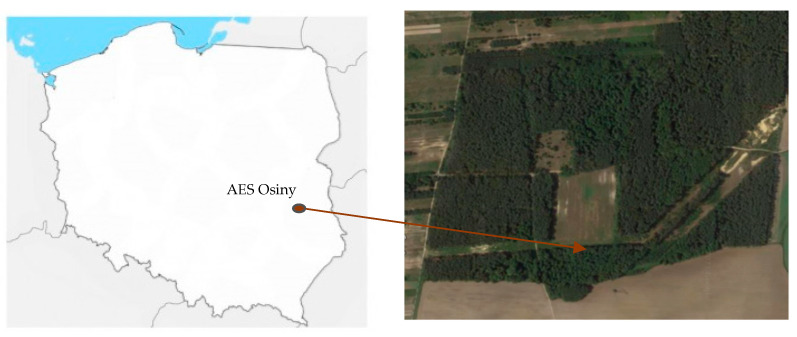
Place of sampling—Agricultural Experimental Station in Osiny, Puławy, Poland.

**Figure 12 ijms-23-02633-f012:**
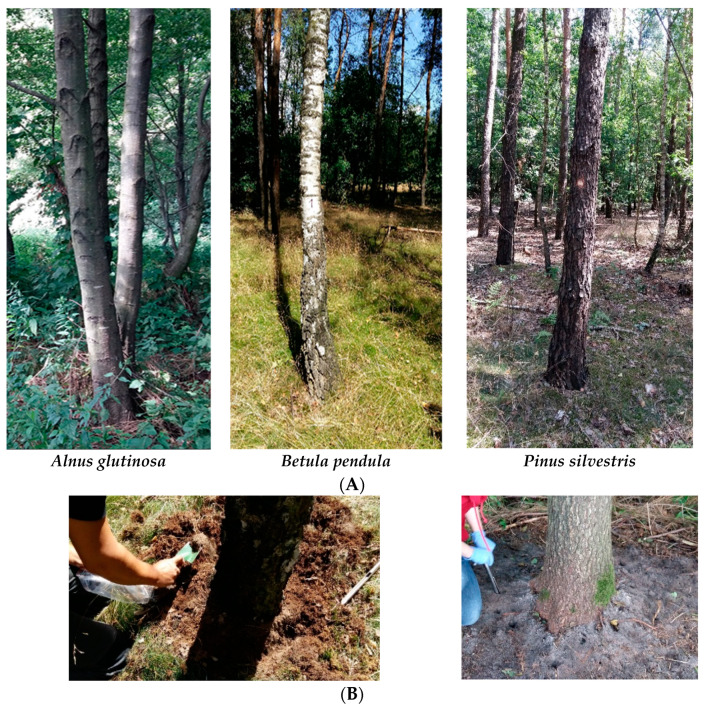
(**A**) Ecosystem of the selected tree species. (**B**) Soil sampling.

**Table 1 ijms-23-02633-t001:** Classification and basic characteristic of soil samples collected from peri-root zone of trees in summer 2019 and 2020.

Tree	Symbol	Soil Texture Class (USDA) *	Particle Size Distribution [%]			C_org_ (%)	Humus (%)	C_tot_ (%)	N_tot_ (%)	C/N	P	K	Mg	pH
			Sand (2.0–0.05 mm)	Silt (0.05–0.002 mm)	Clay (<0.002 mm)									
*Betula pendula*	Bp1	sand	96.62	3.36	0.02	1.830	3.155	2.121	0.173	12.260	6.979	6.354	1.640	3.235
	Bp2	sand	95.68	4.12	0.19	1.376	2.372	1.587	0.127	12.492	5.532	4.907	0.890	3.225
	Bp3	sand	95.73	4.17	0.09	1.357	2.339	1.536	0.131	11.770	10.192	5.692	1.330	3.190
*Alnus glutinosa*	Ag1	sand	91.85	8.00	0.16	0.988	1.703	1.133	0.065	17.502	3.183	1.290	0.380	3.500
	Ag2	sand	93.52	6.45	0.02	1.093	1.884	1.236	0.071	17.473	6.170	1.014	0.300	3.190
	Ag3	sand	92.98	7.00	0.02	0.640	1.103	0.716	0.039	18.346	5.478	1.086	0.350	3.710
*Pinus sylvestris*	Ps1	sand	95.58	4.40	0.02	1.157	1.994	1.284	0.080	16.044	1.115	1.553	0.810	3.080
	Ps2	sand	92.94	6.54	0.52	0.999	1.723	1.117	0.068	16.419	1.458	1.280	0.400	3.020
	Ps3	sand	93.64	5.86	0.50	1.251	2.157	1.422	0.088	16.251	1.352	1.643	0.280	2.985

* Determined on the basis of the United States Department of Agriculture (USDA) Natural Resources Conservation Service; Soil Taxonomy A Basic System of Soil Classification for Making and Interpreting Soil Surveys.

**Table 2 ijms-23-02633-t002:** Enzymatic activities in soil samples collected from peri-root zone of tree: silver birch (Bp; *Betula pendula*), black alder (Ag, *Alnus glutinosa*), and Scots pine (Ps, *Pinus sylvestris*).

Sample	DHA	AlP	AcP
**2019**			
**Bp1**	6.283 ^b^ ± 0.705	9.535 ^b^ ± 0.298	41.633 ^b^ ± 2.033
**Bp2**	8.044 ^b^ ± 0.806	9.337 ^b^ ± 0.346	41.442 ^b^ ± 3.828
**Bp3**	5.262 b ± 0.305	8.611 ^b^ ± 0.704	35.007 ^b^ ± 0.388
**Ag1**	51.504 ^a^ ± 2.618	16.113 ^a^ ± 0.396	96.383 ^a^ ± 2.454
**Ag2**	35.676 ^a^ ± 2.835	14.667 ^a^ ± 0.409	79.701 ^a^ ± 2.715
**Ag3**	26.270 ^a^ ± 1.504	13.359 ^a^ ± 0.280	82.302 ^a^ ± 6.598
**Ps1**	0.931 ^c^ ± 0.115	7.874 ^c^ ± 0.856	16.934 ^c^ ± 1.614
**Ps2**	1.151 ^c^ ± 0.119	6.591 ^c^ ± 0.357	18.471 ^c^ ± 1.016
**Ps3**	0.607 ^c^ ± 0.005	6.670 ^c^ ± 0.417	13.778 ^c^ ± 1.120
**2020**			
**Bp1**	12.064 ^b^ ± 0.649	11.513 ^b^ ± 1.089	38.436 ^b^ ± 2.565
**Bp2**	12.027 ^b^ ± 0.617	10.585 ^b^ ± 1.300	44.292 ^b^ ± 1.347
**Bp3**	9.473 ^b^ ± 0.725	10.142 ^b^ ± 0.269	37.711 ^b^ ± 2.937
**Ag1**	25.067 ^c^ ± 0.427	15.077 ^a^ ± 0.697	81.241 ^a^ ± 5.076
**Ag2**	19.952 ^c^ ± 0.320	14.232 ^a^ ± 0.136	80.230 ^a^ ± 1.743
**Ag3**	24.525 ^c^ ± 0.206	13.109 ^a^ ± 0.443	74.925 ^a^ ± 1.951
**Ps1**	1.875 ^c^ ± 0.118	7.085 ^c^ ± 0.567	16.252 ^c^ ± 0.695
**Ps2**	1.145 ^c^ ± 0.118	7.740 ^c^ ± 0.256	15.009 ^c^ ± 1.023
**Ps3**	1.377 ^c^ ± 0.061	6.704 ^c^ ± 0.211	14.571 ^c^ ± 1.567

DHA—dehydrogenases (μg formazan/g dry matter (d.m.) of soil/24 h); AlP—alkaline phosphatase (μg p-nitrophenol/g d.m. of soil/h); AcP—acid phosphatase (μg p-nitrophenol/g d.m. of soil/h). Treatment means with different letters are significantly different (Tukey’s mean separation test, *p* < 0.05; *n* = 3).

**Table 3 ijms-23-02633-t003:** Changes in microorganism metabolic diversity in bulk soils from the peri-root zone of trees as evaluated by Shannon’s general diversity index (*H*’**), substrate richness (R), substrate evenness (E), and average well-color development (AWCD_590_) obtained in the Biolog EcoPlates incubated for 120 h. Treatment means separated by different letters are significantly different (Tukey’s mean separation test, *p* < 0.05); *n* = 3.

Sample ID	*H’*	R	E	AWCD_590_
**2019**				
**Bp1**	3.148 ^b^ ± 0.016	26.000 ^b^ ± 0.577	0.986 ^b^ ± 0.005	1.342 ^b^ ± 0.057
**Bp2**	3.052 ^b^ ± 0.031	23.333 ^b^ ± 0.377	0.979 ^b^ ± 0.006	1.345 ^b^ ± 0.133
**Bp3**	3.122 ^b^ ± 0.028	23.000 ^b^ ± 1.732	0.975 ^b^ ± 0.018	1.310 ^b^ ± 0.069
**Ag1**	3.355 ^a^ ± 0.021	30.000 ^a^ ± 0.377	0.957 _c_ ± 0.006	1.873 ^a^ ± 0.048
**Ag2**	3.374 ^a^ ± 0.013	30.333 ^a^ ± 0.577	0.959 ^c^ ± 0.002	1.997 ^a^ ± 0.080
**Ag3**	3.379 ^a^ ± 0.013	30.333 ^a^ ± 0.577	0.960 ^c^ ± 0.002	1.935 ^a^ ± 0.060
**Ps1**	2.609 ^c^ ± 0.033	16.333 ^c^ ± 1.082	1.045 ^a^ ± 0.039	0.424 ^c^ ± 0.030
**Ps2**	2.735 ^c^ ± 0.003	18.333 ^c^ ± 0.577	0.992 ^a^ ± 0.009	0.763 ^c^ ± 0.036
**Ps3**	2.921 ^b^ ± 1.128	17.000 ^c^ ± 1.000	1.032 ^a^ ± 0.049	0.403 ^c^ ± 0.064
**2020**				
**Bp1**	3.144 ^b^ ± 0.025	25.333 ^b^ ± 1.155	0.973 ^b^ ± 0.010	1.338 ^b^ ± 0.061
**Bp2**	3.138 ^b^ ± 0.042	22.000 ^b^ ± 1.732	0.984 ^b^ ± 0.012	1.363 ^b^ ± 0.084
**Bp3**	2.993 ^b^ ± 0.010	23.667 ^b^ ± 0.577	0.988 ^b^ ± 0.006	1.278 ^b^ ± 0.072
**Ag1**	3.361 ^a^ ± 0.019	29.667 ^a^ ± 0.577	0.962 ^c^ ± 0.006	1.792 ^a^ ± 0.173
**Ag2**	3.292 ^a^ ± 0.006	28.333 ^a^ ± 0.577	0.954 ^c^ ± 0.006	1.752 ^a^ ± 0.021
**Ag3**	3.344 ^a^ ± 0.019	29.667 ^a^ ± 0.577	0.977 ^c^ ± 0.001	1.832 ^a^ ± 0.075
**Ps1**	2.607 ^c^ ± 0.026	16.667 ^c^ ± 1.528	0.992 ^a^ ± 0.012	0.539 ^c^ ± 0.055
**Ps2**	2.740 ^c^ ± 0.115	15.667 ^c^ ± 0.528	0.998 ^a^ ± 0.066	0.776 ^c^ ± 0.076
**Ps3**	3.005 ^b^ ± 0.031	18.333 ^c^ ± 1.528	0.964 ^a^ ± 0.012	0.660 ^c^ ± 0.039

**Table 4 ijms-23-02633-t004:** Number of amplicon sequence variants detected in each sample.

Sample ID	ASVs	Classified ASVs (%)	Sequence Read Archive (SRA) NCBI ID
**2019**			
**Bp1**	100	59	SRR16926273
**Bp2**	123	62	SRR16926272
**Bp3**	88	59	SRR16926271
**Ag1**	227	66	SRR16926270
**Ag2**	183	64	SRR16926269
**Ag3**	187	64	SRR16926267
**Ps1**	87	62	SRR16926266
**Ps2**	79	61	SRR16926265
**Ps3**	178	65	SRR16926264
**2020**			
**Bp1**	106	55	SPR16962104
**Bp2**	106	57	SPR16962103
**Bp3**	111	57	SPR16962102
**Ag1**	172	65	SPR16962101
**Ag2**	174	61	SPR16962100
**Ag3**	225	65	SPR16962098
**Ps1**	85	60	SPR16962097
**Ps2**	105	64	SPR16962096
**Ps3**	46	52	SPR16962095

## Data Availability

Not applicable.

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
