# Peer review of "Biodiversity and Metabolic Potential of Bacteria in Bulk Soil from the Peri-Root Zone of Black Alder (Alnus glutinosa), Silver Birch (Betula pendula) and Scots Pine (Pinus sylvestris)"

_ijms, 2022, doi:10.3390/ijms23052633_

Round 1

Reviewer 1 Report

This work examines the traits of enzymatic activities and bacterial diversity linked to the soil health of tree populations. The study is interesting and quite complete, however, it would be convenient to give it a less descriptive point of view and include the functionality of the results as a quality and health control tool in the soils of forested areas or forest production. Layout is adequate, but requires some text revisions and organization. The figures are out of square and sometimes require a little more explanation. The work is very important in terms of forest management and involves a novel and profound approach, which gives it strength. The studies linked to this will suppose a substantial improvement in the ecosystem and use of the forest land.

Abstract
-Please take out this information about methodologies that are not completely necessary here
-Results resume and explanation requires a better linkage more than an enumeration in the abstract
-Maybe include some conclusion line will help with the final aim of the abstract

Intro
-I would include a reason for the three tree species selection. Just is maybe local relevance or their representativeness will be ok, but as the authors include in this work, just seems a random choice

Results
-Soil description: seems like enumerating factors or characteristics. It will help a lot in the reading and understanding linking them in a succession, indication the importance. Alternatively, this section could be reduced just indicating that Table 1 has the full characterization
-2019 and 2020 differences could be not only exposed, but explained in a climate condition, year variation factor or similar.
-Explanation of Figure 2 result is just a description that doesn't contribute more than in materials and methods section. This could be improved to help in the results interpretation
-Explanation of PCA analysis is not well addressed, or the explanation is not fully developed. Some conclusions arr not well reflected in the graphs. Please, have a look
-Some data here explained belongs to Material and Methods section 
-Dendrogram result is not completely consistent with the explained results

Figures
-Most of the graphs (bars, PCA...) and some other figures need to curate their highlighted areas, number and code positions and even some labels. Some conclusions are a little difficult to understand is only attend the figures 
-In the case of the microbiota graphs, the resolution is very low, just take into the account for edition issues
-Some labels on figure are not described in figure captions

Materials and Methods
-Check some names (they include some wording fails)

Discussion
-First 3 paragraphs are basically an Introduction again

Conclusions
-Despite they are correct, correlation of microbial diversity and enzymatic activity with soil characteristic should be better addressed (maybe in Discussion). Year variations, also need to be better reflected in the text

Reviewer 2 Report

MAJOR COMMENTS

My main doubt in relation to this work is that the authors continually talk about rhizospheric soil, when from the way of sampling (as shown in Figure 12B) it cannot be concluded that the authors have sampled rhizospheric soil. To do so, they would have had to dig around the tree until they could access the root system and take soil samples in direct physical contact (rhizospheric soil is that layer of soil a few mm wide, depending on the plant species, that is in direct physical contact with the root). Instead, it appears from Figure 12B that they have sampled bulk soil. This point is critical to be able to accept the work in its current format. Otherwise the authors would have to talk about bulk soil taken in the proximity of the root system, but they could not talk about rhizospheric soil samples.

Also grammar and style of language should be reviewed all along the text

MINOR COMMENTS.

-- page 2; line 87. The term “Metatranscriptomic” should be changed to “Metagenomic”, that is the authors have carried out in this work. Metatranscriptomics is the science that studies gene expression of microbes within natural environments, and therefore RNA is extracted from soil, which is later copied into DNA which is then sequenced, indicating the microorganisms that are metabolically active in a particular ecological niche.

-- Page 3. Lines 98-102. Again the authors talk about “bacteria inhabiting the rhizosphere” when in my opinion they have analyzed bulk soil in the proximity of root systems, which is totally different. Also the established hypothesis is wrong since the authors talk about “the microbiome” of the 3 tree species analyzed. They have not analyzed the microbiome of the trees, since they have not analyzed microbial population on other parts of the plant (leaves, bark, branches, endomicrobiome,…..) but only the composition of microbial populations in soil. In fact in page 3/lines 108-109 the authors clearly indicate: “Samples were taken from tree root layers (0–20 cm) 108 according to the Polish Standard”.

-- Page 3 / Table 1. Please, modify the width of the eight column (Humus), so that the word “Humus” is not cut in 2 lines.

-- Page 3 / Lines127-128. Please, change that sentence to “All investigated soil samples exhibited a clear acidic pH”.

-- Page 3 / last line and first lanes of page 4 (Table footnote). The table footnote should be removed since its describing some methodology that should be described in the Materials and Methods section.

-- Pages 5-6/ Fig 1. Obviously the boxes are not in the right place. Please correct the figure.

-- Page 5. Line. 154. Please correct “The utilization per cent….”

-- Page 6 / Legend of Figure 1. Please change the text to: “Percentages of use of selected groups of…..”.

-- Page 6 / lines 172-173. Please, rewrite the sentence to: “The compounds most widely metabolized in the A. glutinosa rhizosphere were…….”

-- Page 10 /lIne 232: Acinobacteria?????. I suppose you want to mean Acidobacteria.

-- Page 10 /line 232-236 and 244-245 in page 11. The exact percentage for every taxon should be mentioned in the text, since it is difficult to deduce the date from Figure 6.

-- Page 12. Figure 7 should be removed since does not contribute anything significant to the work

Author Response

Reviewer 2

Thank you for your review and valuable comments. The authors of the paper took into account all the comments of the reviewer in the text. below I am sending answers to individual questions. All changes to the text are marked in red.

We declare, that the paper has not been published previously elsewhere nor is being considered by another journal, in any language. Also we confirm that the paper has been submitted to Proof-Reading-Service.com for editing and proofreading. We enclose a language proofreading certificate. All authors have seen and agreed to the submitted version. Authors declare no conflict of interest.

We agree with the comment of the reviewer regarding the name of the samples sampling. The authors correctly provided the description of the sampling, but incorrectly named it the rhizosphere. As the reviewers rightly noted, the authors collected bulk soil from peri-zone of selected tree. The description of sampling has been changed throughout the paper. This is a major change throughout the text.

Abstract and title were modified.

The introduction has also been changed to provide the correct hypothesis. Tables and figures were reformatted according to the journal's requirements. Moreover, all minor remarks of the reviewer in the text were taken into account.

Reviewer 3 Report

The manuscript of Anna Gałązkaet al is devoted to the study of the microbial profile and metabolic potential of the rhizosphere of three different trees.
Overall, the authors presented interesting data on microbial profiling in relation to tree species, based on a two-year study.
However, there are several points that raise questions.
1. Title of the manuscript. I doubt that the main results refer to the molecular interactions of bacteria in the rhizosphere. This part of the title raises a question and, it seems to me, requires a paraphrase.
2. The data presented in table 1 do not quite correspond to their description. Thus, the data on the content of Mg in the samples of Pinus sylvestris differ by almost three times. It seems to me that this does not allow us to conclude that these indicators for A. gputinosa are the lowest. The presented results only allow us to conclude that they were the highest for B. pendula (lines 125). And that, if we do not take into account that for P. sylvestris and B. pendula one of the values was practically comparable. Also noteworthy are the data on the distribution of particles in Clay, where the values vary within 25 times.
3.  In my opinion, the third set of data that raise questions are the enzymatic activity data. In this regard, it requires clarification why acid and alkaline phosphatases were chosen to determine the activity, the optimum pH of which is 6.5 and 11, respectively, if the pH values of all the studied soils did not exceed 3.5, and in general were even lower. What data did the authors want to obtain when conducting these tests? From the presented data, it can be seen that for alkaline phosphatase, the data are at a low level. What values of enzyme activity determined by this test are reliably significant? For acid phosphatase, it can still be assumed that, since the pH optimum of this enzyme is closer to soil pH values, the higher activity values can be explained. However, I doubt that enzymes with an optimum at 7 and 11 would be active at pH around 3. It would be helpful if the authors cited similar results from other researchers, if any, for this soil type.

Another small question - line 232 - Acinobacteria - what is it? Probably acidobacteria?
In general, the MS presents interesting results and may be published after corrections have been made. 

Author Response

Reviewer 3

Thank you for your review and valuable comments. The authors of the paper took into account all the comments of the reviewer in the text. All changes to the text are marked in red.

We declare, that the paper has not been published previously elsewhere nor is being considered by another journal, in any language. Also we confirm that the paper has been submitted to Proof-Reading-Service.com for editing and proofreading. We enclose a language proofreading certificate. All authors have seen and agreed to the submitted version. Authors declare no conflict of interest.

We agree with the comment of the reviewer regarding the name of the samples sampling. The authors correctly provided the description of the sampling, but incorrectly named it the rhizosphere. As the reviewers rightly noted, the authors collected bulk soil from peri-zone of selected tree. The description of sampling has been changed throughout the paper. This is a major change throughout the text.

Abstract and title were modified.

The introduction has also been changed to provide the correct hypothesis. Tables and figures were reformatted according to the journal's requirements. Moreover, all minor remarks of the reviewer in the text were taken into account.

The text also includes a description of why both phosphatases were tested.

Determination of DHA, AcP and AlP in the soil samples give us large amount of infor-mation about biological characteristic of the soil. It was confirmed that although oxygen and other electron acceptors can be utilized by dehydrogenases, most of the enzyme is produced by anaerobic microorganisms. The activity of acid phosphatase is related to the presence of soil microorganisms inhabiting a given soil. On the other hand, the activity of alkaline phosphatase is related to plant enzymes.

Round 2

Reviewer 2 Report

All the issues indicated in my previous review have been correctly addressed so I can recommend publication of the manuscript in its current reviewed version

Author Response

Thank you for accepting the paper. Your comments have certainly improved our manuscript. 

Reviewer 3 Report

The authors have made some corrections, but the manuscript is still in need of revision.

Lines 148-153. to explain the possible role of enzymes, the authors added several sentences. Reference should be made to each of these points.

However, the authors did not provide an explanation why the activity of alkaline phosphatase was measured in extremely acidic soil. Probably, the authors should somehow discuss the low activity of alkaline phosphatase precisely because of the absolutely non-optimal pH values. And provide citation on the values ​​of alkaline phosphatase activity in soils with a different pH value.

Line 409 - The authors should note that only acid phosphatase showed relatively high activity in the samples taken from B.p. and A.g. This conclusion does not apply to alkaline phosphatase. It is this place that requires comparative data on the activity of enzymes from the literature.

There are some typos

line 27 - double "from"

authors should correct the format of the borders in table 2

Author Response

Dear Reviewer 3

Thank you for your additional comments and remarks. The authors tried to make all the required corrections to improve the paper. All changes to the text are marked in blue.

We have completed the information regarding enzyme studies. However, these studies are only part of the work that mainly focuses on the assessment of structural and functional biodiversity. Nevertheless, soil enzyme studies are very important. In many studies, regardless of the soil reaction, the activity of various enzymes, including acid and alkaline phosphatase, is determined in parallel. We cited in the text both research by other authors and our own in this area. The activity of acid phosphatase is related to the presence of soil microorganisms in-habiting a given soil. On the other hand, the activity of alkaline phosphatase is related to plant enzymes. In our research, we evalueted acidic forest bulk soils that are not closely related to tree roots, hence we showed a much lower activity of alkaline phosphatase compared to acid phosphatase activity.

In line with the reviewer's comments, minor corrections were also introduced:

line 27, the double "from" was removed,

table 2 was reformatted in accordance with the publishing requirements,

In line with the reviewer's comments, minor corrections were also introduced:

line 27, the double "from" was removed,

table 2 was reformatted in accordance with the publishing requirements.

new literature has been added and the numbering in the list of references has been changed..

We declare, that the paper has not been published previously elsewhere nor is being considered by another journal, in any language. Also we confirm that the paper has been submitted to Proof-Reading-Service.com for editing and proofreading. We enclose a language proofreading certificate. All authors have seen and agreed to the submitted version. Authors declare no conflict of interest.